# The Effects of Pre-Game Carbohydrate Intake on Running Performance and Substrate Utilisation during Simulated Gaelic Football Match Play

**DOI:** 10.3390/nu13051392

**Published:** 2021-04-21

**Authors:** Luke O’Brien, Kieran Collins, Richard Webb, Ian Davies, Dominic Doran, Farzad Amirabdollahian

**Affiliations:** 1School of Health Sciences, Liverpool Hope University, Liverpool L16 9JD, UK; webbr1@hope.ac.uk (R.W.); amirabf@hope.ac.uk (F.A.); 2Gaelic Sports Research Centre, Technological University of Dublin, Tallaght, D24 FKT9 Dublin, Ireland; Kieran.Collins@tudublin.ie (K.C.); D.A.Doran@ljmu.ac.uk (D.D.); 3Research Institute of Sport and Exercise Science, Liverpool John Moores University, Liverpool L3 5AF, UK; I.G.Davies@ljmu.ac.uk

**Keywords:** Gaelic games, carbohydrate, running performance, substrate utilisation

## Abstract

Background: Previous research has reported that elite Gaelic football players’ carbohydrate (CHO) intakes are sub-optimal, especially, in the lead up to competitive matches. Despite clear decrements in running performance across elite Gaelic football matches, there are no studies that have investigated nutrition interventions on match-related Gaelic football performance. The aim of this study was to determine whether a higher-CHO diet in line with sports nutrition guidelines can improve Gaelic football-related performance compared to lower CHO intakes previously observed in Gaelic footballers. Methods: Twelve Gaelic football players completed a Gaelic football simulation protocol (GFSP) on two occasions after consuming a high-CHO diet (7 g·kg^−1^) (HCHO) or an energy-matched lower-CHO diet (3.5 g·kg^−1^) (L-CHO) for 48 h. Movement demands and heart rate were measured using portable global positioning systems devices. Countermovement jump height (CMJ) and repeated-sprint ability (RSA) were measured throughout each trial. Expired respiratory gases were collected throughout the trial using a portable gas analyser. Blood samples were taken at rest, half-time, and post-simulation. Results: There was no significant difference in total distance (*p* = 0.811; η^2^ = 0.005) or high-speed running distance (HSRD) covered between both trials. However, in the second half of the HCHO trial, HSRD was significantly greater compared to the second half of the LCHO trial (*p* = 0.015). Sprint distance covered during GFSP was significantly greater in HCHO (8.1 ± 3.5 m·min^−1^) compared with LCHO (6.4 ± 3.2 m·min^−1^) (*p* = 0.011; η^2^ = 0.445). RSA performance (*p* < 0.0001; η^2^ = 0.735) and lower body power (CMJ) (*p* < 0.0001; η^2^ = 0.683) were significantly greater during the HCHO trial compared to LCHO. Overall CHO oxidation rates were significantly greater under HCHO conditions compared to LCHO (3.3 ± 0.5 vs. 2.7 ± 0.6 g·min^−1^) (*p* < 0.001; η^2^ = 0.798). Blood lactate concentrations were significantly higher during HCHO trial versus LCHO (*p* = 0.026; η^2^ = 0.375). There were no significant differences in plasma glucose, non-esterified fatty acids (NEFAs), and glycerol concentration between trials. In both trials, all blood metabolites were significantly elevated at half-time and post-trial compared to pre-trial. Conclusion: These findings indicate that a higher-CHO diet can reduce declines in physical performance during simulated Gaelic football match play.

## 1. Introduction

The physical demands of elite Gaelic football match play are well documented [1,2,3,4,5]; however, the relative balance of energy sources fuelling the demands of match play are unknown. Therefore, there is a requirement to better understand substrate utilisation during Gaelic match play in order to inform optimal fuelling strategies. Malone et al. (2017a) reported significant reductions in match running performance across both halves and quarters of play [1]. A reduction in running performance has also been reported during elite-level soccer, rugby league and Australian football matches [6,7,8]. Fatigue is complex and its multifactorial occurrence is often attributed to a reduction in glycogen stores, dehydration and reduced muscle pH via increases in hydrogen ions or pacing strategies [6,9,10]. These findings may suggest a need for more efficient pre-game nutrition strategies to maximise glycogen stores and delay fatigue. 

Most intermittent team sport athletes use both anaerobic and aerobic energy systems, which depend on carbohydrate (CHO) as the primary fuel source [11]. Low glycogen concentrations may have a detrimental effect on players’ ability to meet and maintain the physical and physiological demands of match play due to glycogen being the main energy substrate utilised by skeletal muscle during periods of high-intensity activity [11,12]. Low CHO intakes in the days preceding match play may well be a contributing factor to the reported decrements in running metrics, such as total distance (TD), high-speed running distance (HSRD) and sprint distance (SD) achieved during game performance [1]. It is suggested that CHO consumed in line with the guidelines can optimise muscle glycogen stores, which in turn can reduce decrements in running performance across match play [13]. Previous research has reported that elite Gaelic footballers consume CHO intakes of <4 g·kg^−1^.day^−1^ in the days preceding competition [13,14], which is insufficient and will likely result in sub-optimal muscle glycogen concentrations [11]. Therefore, there is a need to investigate what effect CHO intake in line with expert sports nutrition guidelines in the days preceding competition has on running performance and substrate utilisation in Gaelic football players compared a relatively low CHO intake similar to that observed in previous research [13,14].

Understanding of the relative balance of substrate used will help determine pregame macronutrient intake targets to attenuate decrements in game performance. Previous studies that have used gas analysis to assess substrate utilisation demonstrated greater CHO oxidation rates during simulated soccer and squash matches following higher CHO intake [15,16], and these higher CHO oxidation rates resulted in less decrements in physical performance. Similarly, blood metabolites can give a good indication of substrate utilisation during exercise—higher concentrations of plasma glucose and lactate signify greater levels of CHO oxidation [17]. High blood lactate during match play signifies that the rate of glycolysis is high as lactate is an intermediate of the glycolytic system [17]. Bangsbo et al. (2006) demonstrated that blood non-esterified fatty acids (NEFAs) and glycerol concentrations increase progressively during a soccer match, indicating an increase in fat oxidation compensating for the lowering of muscle glycogen. The use of gas analysis and blood sampling during Gaelic football-specific activity may help to determine whether lower CHO intake and oxidation are implicated in fatigue. 

Despite clear decrements in the running performance of elite Gaelic football players across matches (Malone et al., 2017a), no studies have examined the effect of pre-game nutrition interventions to improve match-related running performance and delay fatigue in Gaelic footballers. The aim of the present study was to assess the match-related running performance and substrate utilisation during simulated Gaelic football match play following 48 h of high (7 g·kg^−1^) or low (3.5 g·kg^−1^) CHO availability. We hypothesise that (i) consuming 7 g·kg^−1^ CHO would improve high-intensity running performance and delay the onset of fatigue better compared with the 3.5 g·kg^−1^ CHO diet and (ii), the higher CHO intake would result in greater rates of CHO oxidation during simulated match play.

## 2. Materials and Methods

### 2.1. Participants

Twelve male Gaelic football players (mean age ± SD: 23.3 ± 2.1 years, stature: 1.83 ± 0.06 m, body mass: 79.8 ± 7.5 kg, VO_2peak_: 50.8 ± 3.3 mL.kg^−1^.min^−1^, Peak Power Output (PPO): 302.5 ± 24.9 W) participated in this study. Statistical power was calculated using statistical power analysis software (GPower v3.1, Germany) and a sample size of 8 was estimated using data from Raman et al. (2014) in which changes in CHO oxidation during a simulated squash match was compared under conditions of high and low CHO in the preceding days. Using an α level of 0.05, a power (1 − β) of 0.8 and an effect size of 0.81 for differences in CHO oxidation between conditions. Participants were recreational University level and sub-elite club players. All participants participated in collective Gaelic football training twice a week and participated in regular competitive matches. Ethical approval was granted by the local institutional ethics committee (Ethics approval code S 29-11-18 PA 044). Study information was provided prior to gaining written informed consent from the players before commencing data collection.

### 2.2. Experimental Design

A crossover design approach was adopted in this study. The experiment involved two trials—a high-carbohydrates (HCHO) trial and a low carbohydrates (LCHO) trial. Participants performed a familiarisation trial and two main trials in a single-blind (participant), counter-balanced order. Each trial consisted of participants completing a simulated Gaelic football match following 48 h of a dietary intervention. Prior to undertaking the dietary intervention, each participant completed a glycogen-depleting protocol to ensure that they started each dietary intervention with similar glycogen levels. An overview of trial procedures can be seen in Figure 1. 

### 2.3. Gaelic Football Simulation Protocol 

The Gaelic football simulation protocol (GFSP) was designed to replicate the movement and physiological demands of elite Gaelic football matches. The activity profile of the GFSP is based on several match play studies [1,2,3,4,5,18]. The GFSP follows the same cyclical pattern as the previously developed Hurling simulation protocol (HSP); the reliability of which has previously been confirmed [19]. The HSP was used due to the similar movement pattern of the two sports; however, the HSP was adjusted to match the increased running demands observed during Gaelic football match play. To increase both the TD and HSRD, six Repeated Anaerobic Sprint Tests (RAST) were included throughout the protocol at the beginning, mid-point and end of each half. The RAST consists of six 35 m sprints separated by 10 s of recovery between each sprint and has been reported to be a reliable measure of anaerobic power [20]. The GFSP has demonstrated strong test–retest reliability and participants covered an average relative TD of 116.9 m·min^−1^ and relative HSRD (m; >17 km·h^−1^) of 32.4 m·min^−1^ when completing the GFSP [21], which is comparable to elite competitive match play [1]. The GFSP includes two 36 min halves, separated by a 15 min half-time interval.

### 2.4. Preliminary Measurements and Familiarisation

Participants completed a maximal incremental cycling test on a Lode ergometer (Daum Electronic Premium 8i, Furth, Germany) to determine peak power output (PPO) and peak oxygen uptake (VO_2peak_). PPO was determined to prescribe the exercise intensity for each participant’s glycogen depletion protocol. The protocol began at 150 Watts (W) for 2 min and work rate increased by 30 W every minute until exhaustion [22]. Breath by breath analysis (Metamax 3B, CORTEX Biophysik, Leipzig, Germany) was recorded throughout the test. Following a period of recovery, participants reported to an Astroturf pitch for familiarisation with the simulated GFSP, the equipment and the physical demands of the experiment. Participants completing one-half of the GFSP to familiarise themselves with the movement speeds and patterns of the simulated protocol. Familiarisation also included all physical measures such as countermovement jump (CMJ) and Repeated Anaerobic Sprint Test (RAST).

### 2.5. Glycogen-Depleting Protocol

Participants completed a glycogen depletion protocol to ensure that they started both CHO-controlled dietary interventions with similar muscle glycogen levels. The intermittent cycling protocol described by Taylor et al. (2013) [23] was used for the glycogen-depleting exercise protocol. Following a 5 min warm up at a self-selected intensity, participants commenced cycling at 90% of PPO for 2 min followed immediately by 2 min of active recovery at 50% of PPO. This work to recovery ratio was maintained until the subjects could no longer complete 2 min at 90% PPO. At this stage, exercise intensity was then lowered to 80% PPO and when participants were unable to maintain this intensity, it was lowered to 70% and lastly 60% PPO, with the same work to recovery ratio. The exercise protocol was terminated when the participants were unable to complete 2 min at 60% PPO. During the first trial, the activity pattern and time-to-exhaustion for the protocol were recorded for each participant and replicated during their subsequent trial. 

### 2.6. Main Trial

Participants were assigned to either an HCHO (7 g·kg^−1^·day^−1^; 72% of total energy intake) or LCHO (3.5 g·kg^−1^·day^−1^; 36% of total energy intake) diet for 48 h post-glycogen depleted protocol (see Table 1 and Table 2). Participants BM was recorded and used to prescribe the CHO content of the individuals’ diet. Both diets were energy matched and the energy value of the diet was determined using previously reported energy expenditure data of elite Gaelic footballers in the two days preceding a competitive match [21]. Participants were provided with all food required alongside a meal plan. All participants were given strict instructions to follow the diet post-glycogen depletion protocol and in the 48 h preceding the simulated match. Table 1 shows sample details of the high- and low-CHO diets prescribed for an 80 kg male. Participants were instructed to finish all meals provided and not to consume any foods or liquids not prescribed. Participants received a standardised CHO-rich pre-exercise meal providing 2 g·kg^−1^ CHO, consistent with pre-exercise guidelines, i.e., 1–4 g·kg^−1^ CHO in the 4 h before exercise [24], and this same meal was provided prior to both HCHO and LCHO trials.

After 48 h of the dietary intervention, participants arrived at the Sports Performance Laboratory and adjoining Astroturf pitch to complete the simulated match. Blood samples were collected, and participants performed CMJ pre, half-time and post-simulation. Participant’s urine osmolality was assessed using a handheld osmometer (Osmocheck, Perform Better, Southam, UK) prior to each trial to ensure that they were adequately hydrated. Participants were provided with 5 mL·kg^−1^ of water to drink at half-time to standardise fluid intake and minimise dehydration.

### 2.7. Movement and Physiological Measurements

#### 2.7.1. GPS/Heart Rate

Participants wore a portable global positioning systems device recording at 10 Hz (Optimeye S5, Catapult Innovations, Melbourne, Australia). The portable device was worn inside an elastic vest, positioned across the upper back between the scapulae. Devices were activated 30 min before the start of each trial to allow the acquisition of satellite signal. The devices recorded maximum velocity and distances covered within three specific velocity bands corresponding to: 1–16.9 km·h^−1^ categorised as low-speed running, >17 km·h^−1^ categorised as high-speed running and >22 km·h^−1^ categorised as sprint-running distance. HR (Polar, Oy, Kempele, Finland) was monitored and recorded to GPS units for later download and analysis.

#### 2.7.2. Substrate Utilisation

Breath-by-breath gas analysis was recorded for 8 out of the 12 participants using a portable gas analysis system (Metamax 3B, CORTEX Biophysik, Leipzig, Germany) that was calibrated according to manufacturer’s guidelines prior to each trial. Expired-gas data were collected continuously and logged every second throughout the simulation protocol to determine oxygen uptake and carbon dioxide production. Time point markers were entered into this software to identify the beginning, half-time and the end of simulation. The expired-gas data were recorded into the Metasoft 3.0 software (Cortex Biophysik, Leipzig, Germany). The respiratory exchange ratio was calculated by dividing carbon dioxide production by oxygen uptake. Substrate utilisation was calculated by Metasoft 3.0 software, assuming that protein oxidation during exercise was negligible. 

#### 2.7.3. Repeated-Sprint Ability

At the beginning, mid-way point, and end of each half participants performed a Repeated Anaerobic Sprint Test (RAST) to measure anaerobic power. The test involves six sprints over a 35 m distance, with a 10 s recovery between each sprint. For RAST testing, players commenced each repetition from a standing start with a distance of 0.3 m behind timing gates (Brower Timing, Draper, UT, USA). Similar verbal encouragement was provided throughout each effort and both trials. The anaerobic power (P) in each effort can be calculated using the time to complete each sprint (P = total body mass × distance^2^)/time^3^). The mean power (MP) defined as the mean power among the 6 efforts.

#### 2.7.4. Countermovement Jump

CMJ was measured at 3 time points (pre-trial, half-time and post-trial) to assess lower-body power. CMJ height was assessed using a contact timing mat (Just Jump, Probiotics Inc., Huntsville, Alabama, USA). Participants performed 3 jumps separated by 10 s of passive recovery at each time point. The peak jump height of the three jumps was recorded and used for analysis. The Just jump system provides a reliable (CV = 4.2%) and valid measurement of jump height [25]. 

#### 2.7.5. Blood Collection

Blood samples (10 mL) were taken from a superficial vein in the antecubital fossa of the forearm using standard venepuncture techniques (Vacutainer Systems, BD Biosciences, West Sussex, UK). A blood sample was taken 30 min before exercise commenced, at half-time and immediately post-trial. Samples were collected in vacutainers (EDTA, Nu-Care Products, Bedfordshire, UK), and centrifugation at 2000× *g* for 15 min at 4 °C. Following centrifugation, aliquots of plasma were stored at −80 °C for later analysis.

#### 2.7.6. Blood Analysis 

Plasma glucose, lactate, NEFAs and glycerol were analysed using a Randox Daytona autoanalyser and commercially available kits (Randox Laboratories, Belfast, Northern Ireland). The coefficient of variation for plasma glucose, lactate, non-esterified fatty acids and glycerol was ≤5%. 

### 2.8. Statistical Analysis

All data were reported as the mean ± SD throughout. A Shapiro–Wilk test was used to confirm normality and equality of variance was assessed using Levene’s test. A repeated-measures analysis of variance (ANOVA) (time × condition) was used to determine main effects within and/or between experimental conditions. Where a significant time effect is present, post hoc analysis to determine specific differences were identified using the Bonferroni correction. Partial eta-squared (η^2^) values were calculated. For η^2^ data, thresholds of 0.01, 0.09, and 0.25 were considered small, medium and large, respectively [26]. All data were analysed using SPSS statistical software (Version 24.0, Chicago, IL, USA) with statistical significance set at *p* < 0.05.

## 3. Results

### 3.1. Running Performances 

#### 3.1.1. Total Distance Covered

TD was similar for both conditions. No significant condition (*p* = 0.811; η^2^ = 0.005) or interaction effects (*p* = 0.192; η^2^ = 0.149) were observed, but a time effect was observed (*p* = 0.048; η^2^ = 0.311). A 2.4% (2.9 ± 4.3 m·min^−1^) and 1.0% (1.2 ± 1.8 m·min^−1^) reduction in TD covered was observed in the second half compared to the first half in LCHO (*p* = 0.051) and HCHO (*p* = 0.191), respectively (Figure 2).

#### 3.1.2. High-Speed Distance Covered 

There was a significant main effect for time (*p* = 0.001; η^2^ = 0.0633) and interaction (*p* = 0.026; η^2^ = 0.375). A 12% (4.1 ± 3.1 m·min^−1^) and 6% (1.9 ± 2.2 m·min^−1^) reduction in HSRD covered was observed in the second compared to the first half in LCHO (*p* = 0.001) and HCHO (*p* = 0.015), respectively. In the second half of HCHO, HSRD was significantly greater compared to the second half of LCHO (*p* = 0.015) (Figure 2).

#### 3.1.3. Sprint Distance Covered

There was a significant main effect for condition (*p* = 0.011; η^2^ = 0.445) and time (*p* < 0.001; η^2^ = 0.753). However, there was no interaction effect (*p* = 0.987; η^2^ = 0.0001) for SD covered. The SD covered during GFSP was significantly greater in HCHO (8.1 ± 3.5 m·min^−1^) compared with LCHO (6.4 ± 3.2 m·min^−1^) (*p* = 0.011). In the second half of the HCHO, the SD covered was significantly greater compared to LCHO (6.2 ± 3.5 vs. 4.6 ± 3.7 m·min^−1^) (*p* = 0.011) (Figure 2).

#### 3.1.4. Peak Velocity 

No significant condition (*p* = 0.084; η^2^ = 0.246) or interaction effects (*p* = 0.2281; η^2^ = 0.129) were observed, but a time effect (*p* = 0.005; η^2^ = 0.519) was seen for peak velocity. Results demonstrate that peak velocity was significantly reduced in the second half compared to the first by 5.4% (7.4 ± 0.6 vs. 7.0 ± 0.6 m·s^−1^), and 4.0% (7.5 ± 0.5 vs. 7.2 ± 0.5 m·s^−1^) in LCHO (*p* < 0.001), and HCHO (*p* < 0.001), respectively.

### 3.2. Repeated-Sprint Ability

There was a significant condition (*p* < 0.001; η^2^ = 0.735) and time effect (*p* < 0.001; η^2^ = 0.664) for mean anaerobic power, but no interaction effect (*p* = 0.573; η^2^ = 0.066). Mean anaerobic power was significantly higher at first time point compared to all other time points in both trials. Mean power output was significantly greater in HCHO at time point 2 (*p* = 0.034), 3 (*p* = 0.008), 5 (*p* = 0.008) and 6 (*p* = 0.01), compared to LCHO (Figure 3).

### 3.3. Countermovement Jump

There was a significant main effect for condition (*p* < 0.001; η^2^ = 0.683) and time (*p* = 0.007; η^2^ = 0.36), but there was no interaction effect (*p* = 0.066; η^2^ = 0.219) for CMJ. Peak jump height was significantly higher at half-time in the HCHO compared to LCHO condition (*p* = 0.002). Peak jump height was significantly lower post-trial compared to pre-trial in LCHO (*p* = 0.01); however, there was no difference across time points for HCHO (Figure 4).

### 3.4. Substrate Utilisation 

#### 3.4.1. CHO Oxidation

Analysis of the CHO oxidation data indicated a significant condition (*p* < 0.001; η^2^ = 0.798) and time effect (*p* = 0.01; η^2^ = 0.638) but no interaction effect (*p* = 0.131; η^2^ = 0.295). Overall CHO oxidation rates were significantly greater in HCHO compared to LCHO (3.3 ± 0.5 vs. 2.7 ± 0.6 g·min^−1^) (*p* < 0.001). CHO rates significantly decline across halves in LCHO (*p* = 0.021) but not HCHO (*p* = 0.09) (Figure 5).

#### 3.4.2. Fat Oxidation

There was a significant main effect for condition (*p* = 015; η^2^ = 0.597) and time (*p* = 0.032; η^2^ = 0.506) but no interaction effect (*p* = 0.055; η^2^ = 0.431). The HCHO trial displayed significantly lower overall rates of fat oxidation compared to LCHO (0.19 ± 0.11 vs. 0.42 ± 0.25 g·min^−1^) (*p* = 0.015). Fat oxidation was also observed to increase across halves for both HCHO (*p* = 0.017) and LCHO (*p* = 0.034) conditions (Figure 5).

#### 3.4.3. RER

RER data analysis revealed a significant main condition (*p* = 0.004; η^2^ = 0.721) and time effect (*p* = 0.032; η^2^ = 0.506) but no interaction effect (*p* = 0.140; η^2^ = 0.283). RER values were significantly higher in HCHO compared to LCHO (0.98 ± 0.05 vs. 0.92 ± 0.05) (*p* = 0.004). RER decreased across halves in the HCHO (*p* = 0.019) but not LCHO (*p* = 0.058) trial (Figure 5).

### 3.5. Blood Metabolites

#### 3.5.1. Plasma Glucose

There were no significant differences in plasma glucose concentration between trials (*p* = 0.06; η^2^ = 0.285). However, there were main effects of time (*p* < 0.001; η^2^ = 0.685), blood glucose was significantly elevated at half-time (*p* < 0.001) and post-trial compared to pre-trial (*p* = 0.006), during both conditions (Figure 6A). 

#### 3.5.2. Lactate 

There were main effects of condition (*p* = 0.026; η^2^ = 0.375), time (*p* <0.001; η^2^ = 0.897) and an interaction (*p* = 0.001; η^2^= 0.466) of plasma lactate concentration. Lactate was significantly elevated at half-time (*p* < 0.001; *p* < 0.001) and post-trial compared (*p* < 0.001; *p* < 0.001) to pre-trial, during both HCHO and LCHO conditions. Lactate was also significantly greater at half-time compared to post-trial in HCHO (*p* = 0.03). However, there was no difference in these time points in LCHO (*p* = 0.117). Lactate concentration was significant higher at half-time in the HCHO trial compared to LCHO (*p* < 0.001) (Figure 6B). 

#### 3.5.3. Glycerol 

There was no significant main effect for condition (*p* = 0.752; η^2^ = 0.015) or interaction effect (*p* = 0.52; η^2^ = 0.089) for glycerol concertation. However, there was a significant effect of time (*p* < 0.0001; η^2^ = 0.871). Glycerol was significantly elevated at half-time (*p* = 0.027; *p* = 0.019) and post-trial (*p* = 0.003; *p* = 0.005) compared to pre-trial, during both HCHO and LCHO conditions. Glycerol was also significantly greater post-match compared to half-time in LCHO (*p* = 0.049); however, there was no difference in these time points in HCHO (*p* = 0.07) (Figure 6C). 

#### 3.5.4. NEFAs

There were no significant differences in NEFA concentration between trials (*p* = 0.757; η^2^ = 0.015). However, NEFA concentration was significantly elevated at half-time (*p* = 0.019; *p* = 0.012) and post-trial (*p* = 0.003; *p* = 0.006) compared to pre-trial, during both HCHO and LCHO conditions (Figure 6D). 

### 3.6. Heart Rate

There was a significant main effect for condition (*p* = 0.033; η^2^ = 0.38), but there was no significant main effect for time (*p* = 0.059; η^2^ = 0.217) and no interaction effect (*p* = 0.365; η^2^ = 0.099) for mean HR. Mean HR during HCHO and LCHO was 167 ± 8 b·min^−1^ and 162 ± 11 b·min^−1^, respectively. Peak HR was similar between trials (HCHO: 206 ± 14 b·min^−1^; LCHO: 203 ± 16 b·min^−1^) and at all time points, there was no significant main effect for condition (*p* = 0.353; η^2^ = 0.034), time (*p* = 0.471; η^2^ = 0.079) and or interaction effect (*p* = 0.137; η^2^ = 0.166). 

## 4. Discussion

The main finding of this study was that higher CHO intake in the 2 days preceding simulation match attenuated declines in HSRD, SD and RSA throughout the GFSP—when players consumed the HCHO diet, their HSRD and SD were significantly greater in the second half when compared to LCHO diet. Secondly, a greater rate of CHO utilisation was observed during the match simulation following a high-CHO diet. These observations indicate that high-speed running can be better maintained in the second half in a Gaelic football match simulation when CHO intake is sufficient to support higher rates of CHO oxidation. These findings agree with other studies that have demonstrated that higher CHO intakes are associated with delaying the onset of fatigue during simulated team sport performance [14,27].

TD and HSRD covered during the entire protocol were 0.9% and 1.6%, higher, respectively, during the HCHO trial when compared to LCHO—these differences were not significant. SD covered was significantly higher (25.4%) when players’ CHO intake was high compared to the lower CHO intake. In both trials, a reduction in all distances covered at different speeds was observed in the second half compared with the first half of the game. However, in the second half, players covered significantly higher HSRD and SD in the HCHO compared to the LCHO condition. These findings are important for Gaelic football players as reductions in high-speed running during a Gaelic football match play are observed and often attributed to fatigue [1]. The overall success of elite team sports teams is associated with reduced decrements in high-speed running toward the end of a match compared to less successful teams [6]. 

The reasons for decrements in running performance toward the latter stages of intermittent team sport games is unclear; however, fatigue during prolonged intermittent exercise contributes to reduced muscle glycogen [17]. Moreover, increasing muscle glycogen prior to prolonged intermittent exercise by means of consuming a high-CHO diet improves performance [14]. Balsom et al. (1999) reported that during a 90 min 4-a-side game, players performed 33% more high-intensity exercise following 48 h of a high-CHO diet (7.2 g·kg^−1^) compared with a lower-CHO diet (3.3 g·kg^−1^). Contrastingly, Bradley et al. (2016) reported that higher intakes of CHO (6 g·kg^−1^) compared to lower intakes (3 g·kg^−1^) did not affect glycogen availability, glycogen utilisation, or high-intensity running distances during rugby league match play, and this could be due to rugby players being heavier than other team sport athletes, which can result in high amounts of CHO intake even when consuming 3 g·kg^−1^. Furthermore, rugby union match play includes longer rest periods due to the ball being out of play for substantial amounts of time [28]. The activity profile of Gaelic football is different from soccer and rugby, and therefore there was a need to assess the effects of pre-game CHO on Gaelic football activity. The current study demonstrated that intake of 7 g·kg^−1^ of CHO in the two days preceding Gaelic football match play is an effective strategy that can delay fatigue and improve running performance. 

Furthermore, higher CHO intakes were associated with greater RSA and a greater ability to maintain RSA, in comparison to lower CHO intakes. Higher CHO intakes were found to significantly increase mean anaerobic power during four separate RAST that were incorporated in the GFSP. Success within team sports is reliant on players’ capacity to perform repeated bouts of intense, powerful muscular movements such as sprints, jumping or tackling, with short recovery period in between [29]. In team sport athletes, an increased ability to repeat sprints during match play is associated with competing at a higher level [6,30]. The greater anaerobic power produced during the RSA tasks during the simulated game produced by the higher intakes of CHO may represent a vital benefit for Gaelic football players. Repeated sprints exhaust muscle glycogen stores, which results in a reduction in power output and work rate during match play [30]. There is now clear evidence for consuming a higher-CHO diet (7 g·kg^−1^) in the 2 days prior to Gaelic football competition to ensure sufficient muscle glycogen stores to delay fatigue.

CHO oxidation rates were significantly elevated in the HCHO trial compared to the LCHO trial (3.3 vs. 2.7 g·min^−1^). This suggests the need to maximise glycogen stores prior to Gaelic football match play to maintain performance. Conversely, fat oxidation was significantly higher during the LCHO compared to the HCHO trial. Increased muscle glycogen levels in the higher CHO condition and greater muscle glycogenolysis could explain the difference in fuel utilisation [31]; however, muscle glycogen was not assessed in this study so cannot be confirmed. While the rate of CHO oxidation declined during both trials, the reduction in CHO oxidation across halves was greater in LCHO (16%) than the HCHO (5%). The ability to maintain higher rates of CHO oxidation in the second half of the match simulation in the HCHO condition could explain the capacity to perform higher levels of HSRD and a greater ability to maintain RSA in the latter stages of the match compared to LCHO. There are limitations to using expired-gas analyses to estimate CHO and fat oxidation during high-intensity intermittent exercise. Substate utilisation data should be interpreted with caution as expired-gas analyses require steady-state conditions and may not be appropriate during higher-intensity exercise [32]. 

There was no difference in plasma glucose concentration between trials. Plasma glucose levels were greater at half-time and post-trial, compared to pre-trial, for both dietary conditions. Hypoglycemia was not evident in either trial, which indicates that a pre-match meal containing sufficient CHO is sufficient to top up liver glycogen stores and maintain blood glucose concentrations across Gaelic football match play, irrespective of CHO intake in the preceding days. Increased plasma lactate concentrations observed during the HCHO trial may indicate higher a work rate or increased muscle glycogenolysis [33]. 

Plasma NEFA and glycerol concentrations increased throughout the GFSP. Similar findings have been observed during other intermittent team sports [33]. As levels of muscle glycogen diminish, NEFA and glycerol levels increase during a soccer match with increased values seen at half-time and post-match [33]. Although no muscle biopsies were performed in this study, the blood metabolites and previous research would suggest that after 2 days of consuming a HCHO diet participants had higher muscle glycogen concentrations compared with the lower-CHO diet.

The findings in the current study show that the decline in CMJ performance across the simulation protocol is less under the HCHO condition. Participants during the LCHO trial had significantly reduced CMJ height post-simulation compared to pre-trial, while there was no significant difference in CMJ height pre- to post-trial in the HCHO condition. Furthermore, participant CMJ performance was significantly greater at half-time in the HCO compared to the LCHO condition. Therefore, increased CHO availability pre-match enhances the ability to maintain lower body power.

The present study demonstrated that meeting the CHO requirements in the pre-match meal is not enough to make up for a reduced intake of CHO in the 2 days prior to exercise. Under both conditions’ participants consumed the same standardised pre-match meal of 2 g·kg^−1^. However, this was not enough to prevent declines in HSRD, SD and CMJ height in the LCHO compared to the HCHO trial. Chryssanthopoulos et al. (2004) demonstrated that a high-CHO pre-match meal (2 g·kg^−1^) only increased muscle glycogen concentration by 10% [34]. The current study also accounted for dehydration, as both groups had to consume 3 L of water per day pre-trial, levels of hydration were measured pre-trial and there was no difference between groups. Additionally, players were required to consume 5 mL of water per kg of body weight at half-time of the simulation to prevent dehydration. Therefore, dehydration should not be a factor in the decline of running performance, further supporting the theory that reduction in glycogen concentration is the main factor in the reduction in Gaelic football-specific performance. Prolonged intermittent exercise depletes stores of muscle glycogen [31,35] and consuming appropriate CHO intakes before exercise has been shown to delay fatigue [14,31] and improve exercise performance [35,36]. 

It is not advised that players follow the HCHO diet adopted in this study on a consistent basis, as different days have different demands. Players should increase intake of CHO before and around matches and high-intensity training sessions, whereas when energy demands are lower on rest days and low-intensity training days there is not the same requirement for higher CHO intakes. Long-term high daily CHO consumption is associated with increased body weight, body fat content, poor dental health, and metabolic diseases [37]. Future research should focus on longer-term periodised nutritional interventions to ascertain whether this approach could improve Gaelic football performance through enhanced metabolic adaptions, i.e., more efficient use of energy substrates.

## 5. Conclusions

A diet providing 7 g·kg^−1^ CHO in the 2 days preceding simulated Gaelic football match play enabled Gaelic football players to attenuate declines in HSRD, SD, RSA and CMJ height throughout the match simulation compared with a diet containing only 3.5 g·kg^−1^ of CHO. These improvements in physical performance are supported by significantly greater CHO oxidation rates following the higher intake of CHO. This study provides valuable information for Gaelic footballers and coaches who want to maximise the ability to reduce declines in physical performance. The ability to maintain high-speed running performance is considered important in Gaelic football because it enables teams to be more successful, and therefore Gaelic footballers should avoid lower-CHO (3.5 g·kg^−1^) diets and consume a high-CHO diet (7 g·kg^−1^) in the days preceding competition. 

## Figures and Tables

**Figure 1 nutrients-13-01392-f001:**
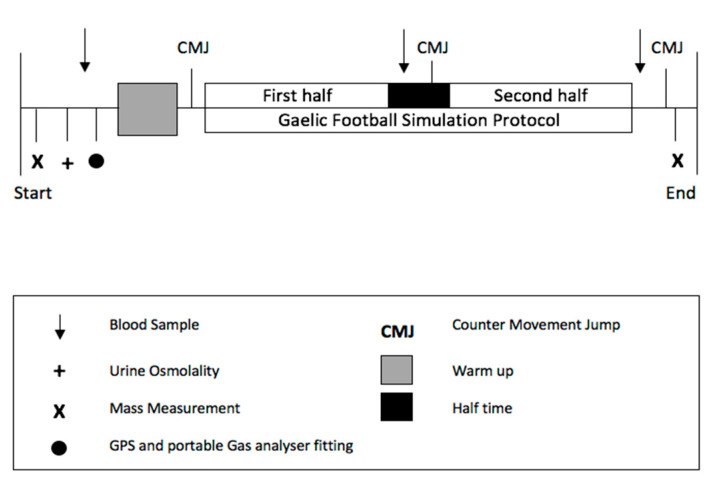
An overview of trial procedures.

**Figure 2 nutrients-13-01392-f002:**
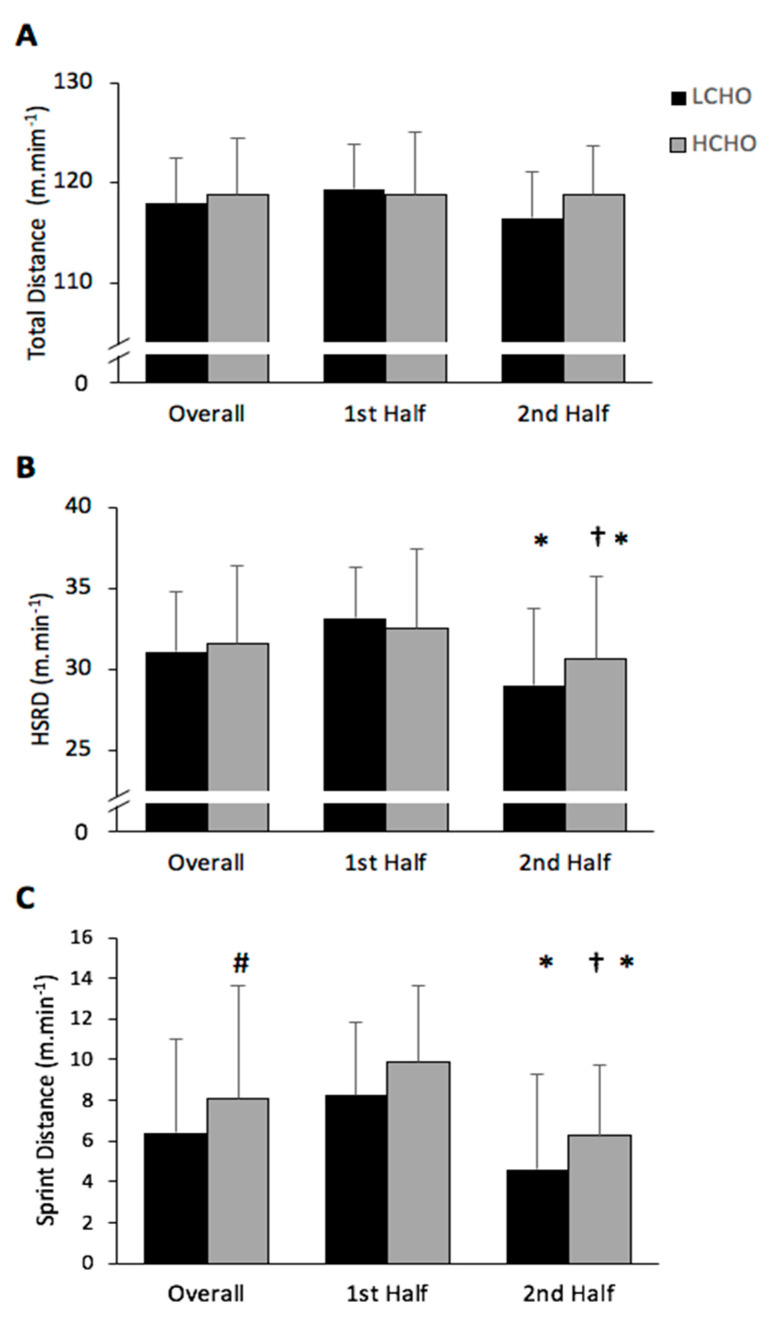
Total distance covered (**A**), high-speed running distance (HSRD) covered (**B**) and sprint distance (**C**) in total and in each half of HCHO and LCHO. # Significant main effect for condition between HCHO and LCHO (*p* < 0.05). * Significant difference from the first half (*p* < 0.05). † Significant different from second half in LCHO (*p* < 0.05). Values are presented as the mean ± SD. HCHO = high-carbohydrate trial; LCHO = low-carbohydrate trial.

**Figure 3 nutrients-13-01392-f003:**
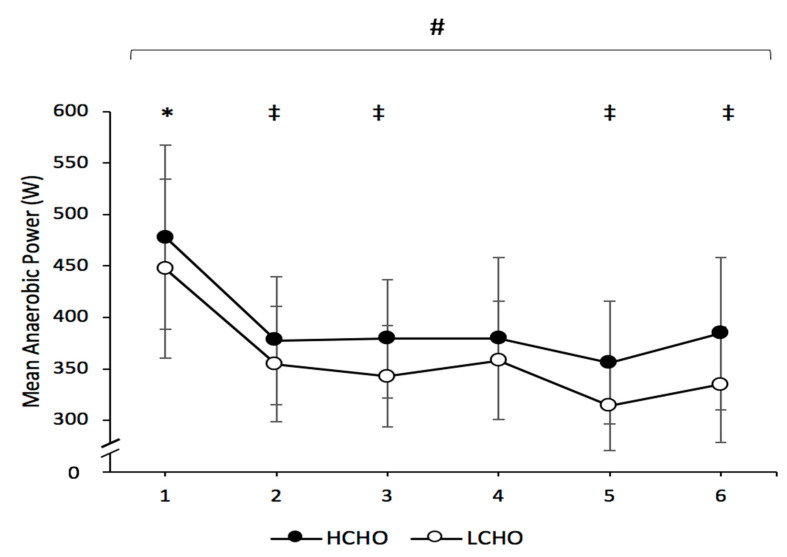
Mean anaerobic power during each Repeated Anaerobic Sprint Test (RAST) throughout both HCHO and LCHO trials. # Significant main effect for condition between HCHO and LCHO (*p* < 0.05). * Significant difference from all other time points in both groups (*p* < 0.05). ‡ Significant difference between HCHO and LCHO at corresponding time point (*p* < 0.05). Values are presented as the mean ± SD. HCHO = high-carbohydrate trial; LCHO = low-carbohydrate trial.

**Figure 4 nutrients-13-01392-f004:**
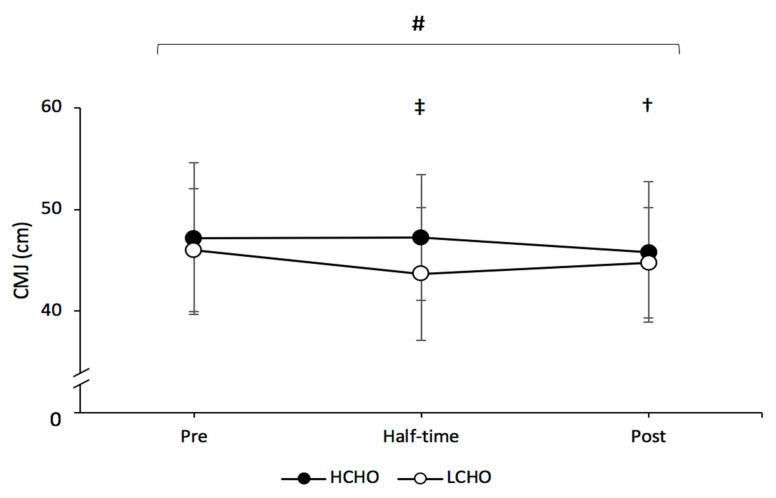
Peak countermovement jump (CMJ) height throughout both HCHO and LCHO trials. # Significant main effect for condition between HCHO and LCHO (*p* < 0.05). ‡ Significant difference between HCHO and LCHO at corresponding time point (*p* < 0.05). † Significantly different compared to pre-trial in LCHO condition (*p* = 0.01). Values are presented as the mean ± SD. HCHO = high-carbohydrate trial; LCHO = low-carbohydrate trial.

**Figure 5 nutrients-13-01392-f005:**
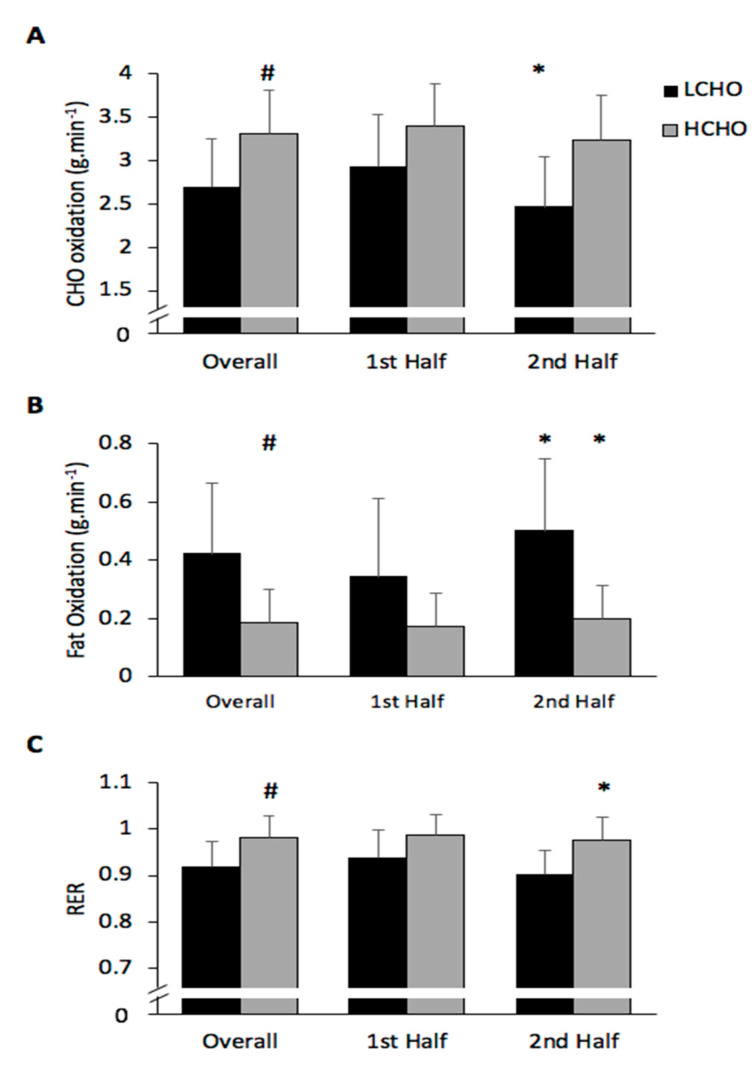
Carbohydrate (CHO) oxidation (**A**), fat oxidation (**B**) and respiratory exchange ratio (RER) (**C**) in total and in each half during HCHO and LCHO trials. # Significant main effect for condition between HCHO and LCHO (*p* < 0.05). * Significant difference from the first half (*p* < 0.05). Values are presented as the mean ± SD. HCHO = high-carbohydrate trial; LCHO = low-carbohydrate trial.

**Figure 6 nutrients-13-01392-f006:**
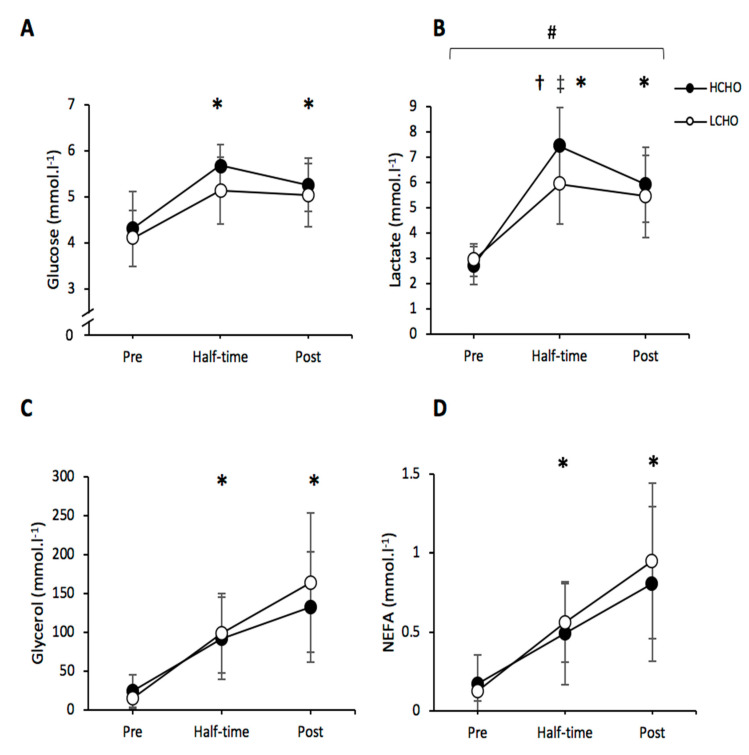
Plasma glucose (**A**), lactate (**B**), glycerol (**C**) and non-esterified fatty acids (NEFAs) (**D**) throughout both HCHO and LCHO trials. # Significant main effect for condition between HCHO and LCHO (*p* < 0.05). * Significant difference from pre-trial in both conditions (*p* < 0.05). ‡ Significant difference between HCHO and LCHO at corresponding time point (*p* < 0.05). † Significantly different compared to post-trial in HCHO condition (*p* < 0.05). Values are presented as the mean ± SD. HCHO = high-carbohydrate trial; LCHO = low-carbohydrate trial.

**Table 1 nutrients-13-01392-t001:** Description of both high- and low-carbohydrate diets prescribed for an 80 kg player.

Time	High-Carbohydrate Diet	Low-Carbohydrate Diet
8.00 a.m.	Cornflakes (100 g)	Cornflakes (50 g)
	Semi skimmed milk (250 mL)	Semi skimmed milk (250 mL)
	Orange juice (250 mL)	2 boiled eggs
	White bread sliced (60 g)	Water (500 mL)
	Strawberry jam (30 g)	
10.30 a.m.	Muller rice (180 g)	Cooked chicken pieces (60 g)
	Banana	1 medium banana
	Lucozade sport (500 mL)	Mixed nuts (60 g)
		Water (500 mL)
1.00 p.m.	Uncle Ben’s basmati rice (250 g)	Uncle Ben’s basmati rice (250 g)
	Chicken breast (90 g)	Chicken breast (90 g)
	Frozen mixed veg (163 g)	Frozen mixed veg (163 g)
	Water (500 mL)	Water (500 mL)
3.30 p.m.	Apple	Greek yogurt (150 g)
	Nutri-grain bar (37 g)	1 medium apple
	Lucozade sport (500 mL)	Water (500 mL)
6.00 p.m.	Pasta (250 g)	Pasta (250 g)
	Domino tomato pasta sauce (125 g)	Cheese (120 g)
	Chicken breast (90 g)	Chicken breast (90 g)
	Frozen mixed veg (163 g)	Frozen mixed veg (163 g)
	Water (500 mL)	Water (500 mL)
9.00 p.m.	Muller rice (180 g)	Beef biltong (65 g)
	Banana	1 medium banana
	Water (500 mL)	Water (500 mL)

**Table 2 nutrients-13-01392-t002:** Energy and macronutrient content of the prescribed diets for 80 kg player.

	High-Carbohydrate Diet (72% of Total Energy Intake)	Low-Carbohydrate Diet (36% of Total Energy Intake)
Energy (kcal/day)	3126	3077
Carbohydrate (g·kg^−1^·day^−1^)	7	3.5
Protein (g·kg^−1^·day^−1^)	1.7	2.8
Fat (g·kg^−1^·day^−1^)	0.5	1.4
Fluid (mL/day)	3000	3000

## Data Availability

The data presented in this study are available on request from the corresponding author.

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
