# Peer review of "The Effects of Pre-Game Carbohydrate Intake on Running Performance and Substrate Utilisation during Simulated Gaelic Football Match Play"

_nutrients, 2021, doi:10.3390/nu13051392_

Round 1

Reviewer 1 Report

The aim of the study conducted by O’Brien et al. was to determine if a higher CHO diet in line with sports nutrition guidelines can improve Gaelic football-related performance compared to lower CHO intakes previously observed in Gaelic footballers. In general the manuscript is well written and the methods and results are clear. There are a however, a few edits to the manuscript that should be made prior to publication.

Introduction should do more to set up why this question is being asked. Reading the introduction it seems clear that consuming adequate carbohydrate prior to a match would be beneficial. Perhaps the authors should discuss the popularity of low carbohydrate diets and why athletes who consume them may have compromised performance. There needs to be more to justify the two groups.

Please provide more detail about glycogen depletion protocol

It is unlikely that a counter movement jump would be impacted by glycogen stores. Please provide justification for using this performance outcome.

Please provide rationale for providing treatments the same pre-game meal.

For HSRD results there is no need to say there was no main effect of treatment if a time by treatment interaction was observed. Just report the interaction and explain where is occurred between the treatments.

It is very difficult to understand what the significant markers are supposed to mean in figures and their legends. The figure markers should be the same across all figures to make it simple for the reader to follow. It is unclear what significantly different from corresponding time point. Does that mean it is different between conditions? For time effects suggest using letters for each time point to show which are different.

Unclear why in figure 5 counter movement jump is being compared to pre-trial values and why the main effect of treatment is not shown in the figure.

Recommend not restating the objective of the study in the discussion, just start with primary findings

The discussion can be shortened. There is plenty of data in the literature to show the changes in substrate oxidation when consuming high or low carbohydrate diets. This paragraph along with paragraphs on the circulating blood analytes can be cut down. This is more supportive data, not novel.

Remove the last paragraph in the discussion section. It is not needed. This information can be incorporated earlier in the discussion.

Reviewer 2 Report

In this study the authors propose to compare the effects of a high CHO diet with a low CHO diet in the days before a simulated Gaelic football match play. The results show that high CHO diet attenuates running performance decline and increases the rate of CHO utilization during exercise, when compared with a low CHO diet.

This study is well designed and results are quite relevant because their application could be immediate.

Major considerations

In methodology It is not clear how RER and gas exchange were measured during Gaelic football match-play, it is necessary to explain it in detail.

Minor considerations

Include macronutrients percentage for each aliment in tables 1 and 2

Include gas analysis in Figure 2

Scale in most of figures does not begin at 0, it must be indicated by double slash

Reviewer 3 Report

The manuscript entitled “The effects of pre-game carbohydrate intake on running performance and substrate utilization during simulated Gaelic foot-3 ball match play” (nutrients-1160610) is well written, interestingly and presents well designed protocol and valuable data. The authors were very well planned and thought about the work.

I agree that carbohydrate intake-related area in sport is a fascinating avenue for investigation, and that further high-quality work is required to elucidate the ideal dietary and supplementation strategies for optimal performance.

The work is prepared carefully however, some elements must be corrected and/or enriched to make this work suitable for publication.

Major remarks:

  1. It is difficult to say about the full impact of diet in this work, and it should be further pointed out that all conclusions only concern short-term nutritional intervention.A longer diet allows for a much more extensive proper metabolic adaptation, e.g. to a better use of energy substrates.There is no clear position in this regard at work.

  1. There is no indication in the paper (especially in the discussion) that not only the consumption of carbohydrates itself is a very important factor, but also their type may modulate the physiological response of the athletes' organism.I mean, for example, the glycemic index of high-carbohydrate diets.It has been shown that the specificity of the glycemic index in the HCHO diet may determine e.g.carbohydrate oxidation rate, the contribution of carbohydrate oxidation to energy yield during exercise.Those adaptations may result in enhanced performance in acute intense exercise and/or running performance.It has been written about it in recent years, e.g. DOI: 1080/09637486.2017.1411891;DOI: 10.3390/nu10030370.I believe that mentioning these aspects would enrich the substantive rank of the work.

  1. If the group only trained twice a week, it would be worth including a clear position on the recreational level of this group.

  1. It should be clearly indicated how was the energy expenditure estimated. Furthermore, was it the basis for determining the energy value of the diets?

  1. Providing a proper calculation of the required sample size is advisable.

Other remarks:

  1. Lines 20 – consider correction – “calorie-matched” - “energy-matched”.
  2. Line 34 – add explanation of the NEFA abbreviation.
  3. Throughout the text, correct and standardize the dots between the units – e.g. instead: "g·kg-1.day-1" use “"g·kg-1day-1”.
  4. Line 73 and 76 – change “levels” to “concentrations”.
  5. Lines 78-79 – I suggest to mention that in addition to CHO intake also “utilization/oxidation” is relevant here.
  6. Line 84 – remove double space.
  7. Line 94 – add unit (“302.5 ± 24.9”).
  8. Figure 1 is difficult to read and should be improved qualitatively and substantially.Please clarify what the "glycogen depleting protocol" consisted of. The authors mention this in chapter 2.5.however, this is still unclear and needs to be clarified.
  9. Line 121 –insert missing "·" between m and min.
  10. Line 167 (tables 1 and 2) - For a better follow-up, combine tables 1 and 2 - just add a column in table 1 and insert the LCHO diet.
  11. Standardize the intervals between the hours and the description, as well as the values and units (Tables).
  12. Table 2 – add units - (ml) “Water (500)”.
  13. Line 182 – authors wrote “analysis was recorded for 8 out of the 12 participants” - does this meet the sample size for a reliable measurement of this parameter?
  14. Correct some entries and delays in the result section, e.g. line 246: "1) (p = 0.011). (Figure 3)."
  15. In the descriptions of figures, all abbreviations should be explained.
  16. Line 263 – clarify – “compared” to LCHO?
  17. Line 278, 311 and 333 – remove double space.
  18. Line 306 – remove the dot (“p < 0.001.; η 2 = 0.685”).
  19. Chapter 3.5.2 (line 316) - Correct the numbering of the figure here and below - it should be number 7 (above is already figure 6).
  20. Chapter 3.5.3. – as above (Figure 7d).
  21. Chapter 3.5.4. – as above (Figure 7c).
  22. Line 325 – correct error in the notation in parentheses (“p < 0.0001 0.752; η 2 = 0.871”).
  23. Chapter 3.6. – limit the record to the accuracy of the method, so there should be no decimal points (HR).
  24. Line 338 – Cited data is missing (this figure is not in the text).
  25. Line 462 – remove the unnecessary period before the comma.

Round 2

Reviewer 3 Report

The manuscript has been sufficiently improved.

Author Response

Thank you for taking the time to once again review our manuscript. We are delighted to hear that you feel the manuscript has been sufficiently improved. Thank you for your suggestions and help in improving our manuscript. We have amended all typographical errors and highlighted them using 'track changes'.